

# GPR27 expression correlates with prognosis and tumor progression in gliomas

Changcheng Cai[1,*], Libo Hu[1,*], Ke Wu[2] and Yinggang Liu[1]

[1] Suining Central Hospital, Suining, China
[2] Xichang People's Hospital, Xichang, China
[*] These authors contributed equally to this work.

## ABSTRACT

**Backgrounds**. Glioma is a highly aggressive type of brain tumor, and its prognosis is still poor despite recent progress in treatment strategies. G protein-coupled receptor 27 (GPR27) is a member of the G protein-coupled receptor family and has been reported to be involved in various cellular processes, including tumor progression. Nevertheless, the clinical potential and tumor-related role of GPR27 in glioma remain unknown. Here we aimed to explore the function and role of GPR27 in gliomas.

**Methods**. In the current study, we evaluated the expression and clinical significance of GPR27 in gliomas using data from The Cancer Genome Atlas (TCGA) datasets. We also conducted cellular experiments to evaluate the functional role of GPR27 in glioma cell growth.

**Results**. We found that GPR27 expression level was closely associated with disease status of glioma. Of note, GPR27 was negatively correlated with WHO grade, with grade IV samples showing the lowest GPR27 levels, while grade II samples showed the highest levels. Patients with IDH mutation or 1p/19q co-deletion exhibited higher GPR27 levels. In addition, lower GPR27 levels were correlated with higher death possibilities. In cellular experiments, we confirmed that GPR27 inhibited glioma cell growth.

**Conclusions**. Our results indicate that GPR27 may function as a potential prognostic biomarker and therapeutic target in gliomas. Further studies are needed to illustrate the signaling mechanism and clinical implications of GPR27 in gliomas.

Corresponding author
Yinggang Liu,
yinggangliu@suiningcentralhosp.com

## INTRODUCTION

Gliomas and glioblastomas are types of primary brain tumors that originate from glial cell, which are the supportive cells of the central nervous system (*Omuro & De Angelis, 2013*). Glioma is the most common form of brain malignancies, accounting for about 80% of brain tumors, and they can vary in grade from slow-growing low-grade tumor to aggressive high-grade tumor. Among gliomas, glioblastoma, also known as glioblastoma multiforme (GBM), represents the most aggressive subtype, representing the highest grade of glioma (*Wen & Reardon, 2016*). Glioblastoma is characterized by its rapid growth, infiltrative nature, and resistance to treatment, making it one of the most

challenging cancers to manage. Understanding the basic features and biology of gliomas and glioblastomas is crucial for advancing diagnosis, prognosis, and treatment strategies for these complex brain tumors (*Melin et al., 2017*).

*GPR27*, a member of the G protein-coupled receptor family, is a protein-coding gene located on human chromosome 3q25.1 (*Pillaiyar et al., 2021*). It encodes a 7-transmembrane receptor that expresses in various tissues and is involved in multiple cellular functions including neurotransmission, immune activation, and cellular growth (*Chopra et al., 2020*; *Dupuis et al., 2017*). Recent research has implied that GPR27 might play critical roles during cancer progression, particularly in hepatocellular carcinoma (HCC) (*Wang et al., 2022*) and breast cancer (*Milioli et al., 2017*). Abnormal expression of GPR27 has been observed in these cancers, and it has been recognized to contribute to tumor growth and angiogenesis. For example, *Wang et al. (2022)* discussed function of GPR27 in HCC progression and its potential mechanism of action through MAPK-ERK signaling pathway. The study highlights the importance of GPR27 in HCC, a type of liver cancer, and proposes that GPR27 may promote tumor progression by activating the MAPK/ERK pathway. The article provides insights into the potential molecular mechanisms underlying HCC development and indicates that GPR27 could be a valuable therapeutic target in HCC treatment. However, further study will be needed to fully understand the exact mechanisms by which GPR27 affects cancer development and to explore its potential as a treatment target for cancers.

This study sheds light on expression and clinical relevance of GPR27 in gliomas and provides evidence for its oncogenic role in glioma growth. The findings presented here have the potential to enhance our understanding of underlying mechanism of glioma progression and could contribute to the development of novel therapeutic strategies for this devastating disease.

# METHODS

## In silico data collection and analysis

RNA expression information in the Fragments Per Kilobase per Million (FPKM) format of glioma samples was obtained as previously published by *Ceccarelli et al. (2016)*. The clinic-pathological information of glioma patients was also extracted ($n = 1122$). The difference of various clinicopathological parameters were compared between the high-GPR27 and low-GPR27 expression groups. Regression analyses were used to assess the relationships between GPR27 expression and clinicpathological variables of glioma cases.

Survival information of glioma cases in The Cancer Genome Atlas (TCGA) glioma databases was analyzed. Kaplan–Meier survival as well as multivariate Cox regression analyses was conducted to identify the patients' prognoses according to GPR27 level as well as other clinic-pathological characteristics.

## Establishment of nomogram and calibration curves

We employed the "RMS" package in R to construct the nomogram for predicting possibility of individual survival. The calibration of the nomogram was evaluated through calibration curves.

## Immune infiltration analyses

We evaluated the association between GPR27 expression and immune cell infiltration in gliomas using ssGSEA algorithm available in the "GSVA" R package (*Hänzelmann, Castelo & Guinney, 2013*). This algorithm helped us assess infiltration status of 24 kinds of immune cell types (*Bindea et al., 2013*). We performed Spearman correlation analysis to clarify correlation between GPR27 expression and immune cell infiltration status.

## Cell culture and transfection

U87 and U251 human glioblastoma cells will be obtained from ATCC and cultured in Dulbecco's Modified Eagle's Medium (DMEM) supplemented with 10% FBS and 1% penicillin-streptomycin at 37 °C with 5% CO2. Culturing medium was replaced every two days, and cells were sub-seeded once they reached 70–80% confluence. Transient transfection of pcDNA3.1-GPR27 plasmids (Cat. #66349; Addgene, Watertown, MA, USA). or control pcDNA3.1-vector plasmids (Cat. #138209; Addgene). was achieved with Lipofectamine 3000 based on manufacturer's instructions. Afterwards, cells were incubated to allow for transfection. After the incubation period, the transfection medium will be replaced with fresh complete growth medium to allow for further cell culture or further experimental tests. Each experiment was repeated three times independently.

## Western blotting

Protein expression levels were assessed through Western blot analysis. Proteins were extracted from the U87 and U251 cell lines utilizing RIPA lysis buffer, and their concentrations were subsequently semi-quantified using a BCA protein assay kit provided by Santa Cruz Biotechnology (Dallas, TX, USA). Proteins, in the amount of 20 μg per lane, were resolved *via* 12% SDS-PAGE and subsequently transferred onto PVDF membranes. These membranes were then blocked using 5% BSA sourced from Sigma-Aldrich for 1 h at ambient temperature before the blocking solution was discarded. Subsequently, the membranes were incubated overnight at 4 °C with primary antibodies: anti-GPR27 (PA5-110977, Invitrogen, Cambridge, MA, USA) and anti-GAPDH (MA1-16757; Invitrogen, Cambridge, MA, USA), both diluted at a ratio of 1:2000. Following the primary antibody incubation, the membranes were washed with TBST containing 0.1% Tween. Then, an HRP-conjugated secondary antibody was applied to incubate at room temperature for 1 h. Detection of the proteins of interest was achieved using the ECL Western Blotting substrate kit (Cattegory #ab65623; Abcam). Each experiment was repeated three times independently.

## Cell proliferation assay

Cell viability was gauged using the Cell Counting Kit-8 (CCK-8), which estimates the quantity of live cells in culture based on the production of a colored formazan product. Briefly, transfected cells were seeded at 5000 cells/well. Then, the CCK-8 experiment was done according to the manufacturer's instruction. Briefly, the medium was replaced with CCK-8 reagent, and the cells were incubated for 2 h. The absorbance of the formazan dye, which is proportional to the number of viable cells, was measured spectrophotometrically at 450 nm. The percentage of cell viability or proliferation was calculated by normalizing to the

control group. Results were expressed as mean ± SD to determine statistical significance. Each experiment was repeated three times independently.

## Statistics

All statistical analyses and plots were performed using R (version 4.1.3). Statistical significance was set at $P < 0.05$. An asterisk (*) indicates $P < 0.05$, two asterisks (**) indicate $P < 0.01$, three asterisks (***) indicate $P < 0.001$.

## Ethics

The Ethics Committee of Suining Central Hospital requires no ethic approval and grants an exemption from informed consent for this public-database study.

# RESULTS

## Aberrant GRP27 expression in gliomas

RNA expression data from TCGA datasets suggested that GPR27 expression was closely correlated with disease status of glioma. For example, GPR27 was negatively correlated with WHO grade, on that grade IV samples showed the lowest GPR27 level while grade II samples showed the highest GPR27 level (Fig. 1A, $P < 0.001$). In contrast, patients with IDH mutation (Fig. 1B) or 1p/19q co-deletion (Fig. 1C) exhibited higher GPR27 levels ($P < 0.001$).

## Correlations between GPR27 expression and clinic-pathological characteristics of gliomas

Among the 1,122 cases from TCGA cohort, we excluded those without sufficient matched clinical information and therefore 699 cases were enrolled for further analysis (Table 1). By dividing TCGA cohorts into high-GPR27 group ($n = 350$) and low-GPR27 group ($n = 349$) based on the median value of log2 (FPKM+1) of GPR27, we further analyzed its correlation with clinic-pathological variables (Table 1). Accordingly, lower GPR27 expression was observed in elder patients, higher-grade gliomas, as well as in the specimens with wild-type IDH status or non-codeletion of 1p/19q (all $P < 0.001$). Considering all the above-mentioned variables had been reported to be correlated with patients' prognosis, we were engaged to further explore whether GPR27 can affect the overall survival of glioma patients.

## GPR27 is an independent survival predictor of gliomas

Among the 699 cases, one case was further excluded due to unavailable survival information. Therefore, the other 698 patients were subjected for survival analyses. Kaplan–Meier survival analysis showed that patients with lower GPR27 level had significantly worse prognosis (Fig. 2A, $P < 0.001$). Briefly, the median overall survival time of low-GPR27 group was 26.3 months (95% CI [23.6–34.1] months), while was up to 99.6 months (95% CI [68.4–135.6] months) of high-GPR27 group. In addition to GPR27, several other variables were identified with prognostic significances in univariate analyses. For example, comparing with WHO grade II patients, WHO grade III patients showed a death hazard ratio as 2.967 (95% CI [1.986–4.433], $P < 0.001$) and WHO grade IV patients showed

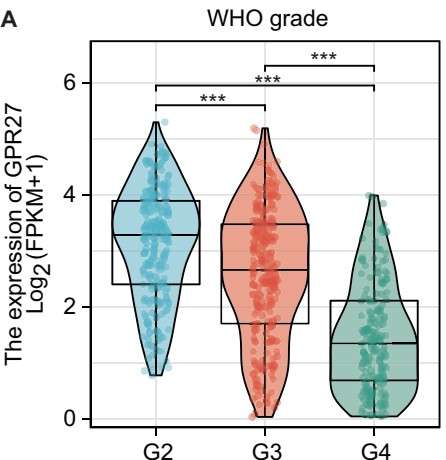

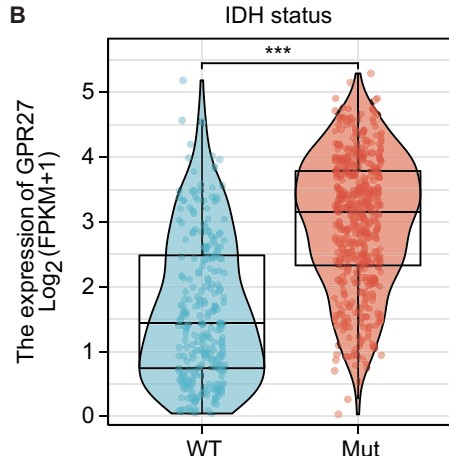

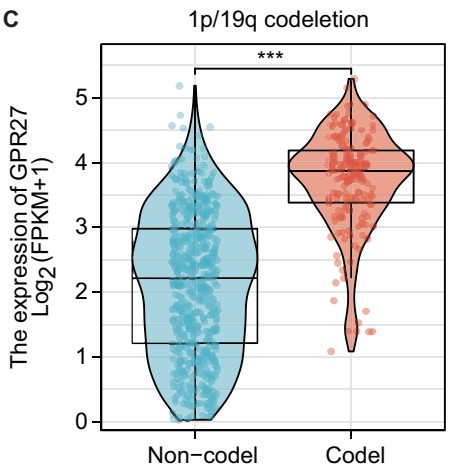

**Figure 1 The correlation between GPR27 expression and disease status in glioma.** (A) GPR27 expression was negatively correlated with WHO grade, with grade IV samples exhibiting the lowest GPR27 levels and grade II samples showing the highest levels ($P < 0.001$). (B) Patients with IDH mutation showed higher GPR27 levels ($P < 0.001$). (C) Similarly, patients with 1p/19q co-deletion exhibited higher GPR27 levels ($P < 0.001$).

a death hazard ratio as 18.6 (95% CI [12.448–27.794], $P < 0.001$). Consistent with our previous data, mutated IDH status indicated a better survival with a death hazard ratio as 0.116 (95% CI [0.089–0.151], $P < 0.001$) comparing to those with wild-type IDH status. Similarly, co-deletion of 1p/19q was a favorable prognostic factor with a death hazard ratio as 0.225 (95% CI [0.147–0.346], $P < 0.001$).

To figure out the independent prognostic factors, we further subjected the variables into a Cox multivariate regression model for survival analysis (Table 2). As a result, elder age and higher WHO grade were confirmed as independent unfavorable prognosis factors. In contrast, mutated IDH status was identified as an independent favorable prognostic factor (hazard ratio 0.265, 95% CI [0.176–0.399], $P < 0.001$). Importantly, our data, for the first

**Table 1  Basic information of enrolled glioma patients.**

| Characteristics | Low GPR27 | High GPR27 | P value |
|---|---|---|---|
| Total cases, n | 349 | 350 | |
| Age, n (%) | | | <0.001[***] |
| ≤60 years old | 245 (35.1%) | 311 (44.5%) | |
| >60 years old | 104 (14.9%) | 39 (5.6%) | |
| Gender, n (%) | | | 0.134 |
| Female | 139 (19.9%) | 159 (22.7%) | |
| Male | 210 (30%) | 191 (27.3%) | |
| WHO grade, n (%) | | | <0.001[***] |
| G2 | 67 (10.5%) | 157 (24.6%) | |
| G3 | 119 (18.7%) | 126 (19.8%) | |
| G4 | 141 (22.1%) | 27 (4.2%) | |
| IDH status, n (%) | | | <0.001[***] |
| WT | 189 (27.4%) | 57 (8.3%) | |
| Mutation | 152 (22.1%) | 291 (42.2%) | |
| 1p/19q codeletion, n (%) | | | <0.001[***] |
| Non-codel | 328 (47.4%) | 192 (27.7%) | |
| Codel | 14 (2%) | 158 (22.8%) | |
| OS event, n (%) | | | <0.001[***] |
| Alive | 162 (23.2%) | 265 (37.9%) | |
| Dead | 187 (26.8%) | 85 (12.2%) | |

**Notes.**
[***]$P<0.001$

time, showed that higher GPR27 expression was an independent benefit biomarker for glioma prognosis (hazard ratio 0.679, 95% CI [0.486–0.947], $P = 0.023$).

Based on the multivariate survival analysis, we also established a nomogram to help predict overall survival of glioma patients (Fig. 2B); the variables in the nomogram included patients' age, gender, WHO grade, IDH status, 1p/19q co-deletion, and GPR27 expression level.

## GPR27 inhibits proliferation of glioma cells

We next conducted western blot experiments to confirm the efficiency of overexpressing GPR27 in U87 and U251 cell lines. Western blot analysis revealed a notable increase in GPR27 levels in cells transfected with pcDNA3.1-GPR27 plasmids compared to cells transfected with the pcDNA3.1-vector. This suggests that the overexpression of GPR27 was successfully achieved (Fig. 3A). To further investigate the role of GPR27 in cell proliferation, we conducted CCK-8 experiments. The results showed that overexpressing GPR27 significantly inhibited the proliferation (Figs. 3B–3C) capacities of both U87 and U251 cell lines. These findings suggested that GPR27 plays a tumor-inhibiting role in glioma cells. Modulation of GPR27 expression may be a promising therapeutic approach for treating glioma. Overall, the results of this study provide important insights into the molecular mechanisms underlying glioma development and progression, and may

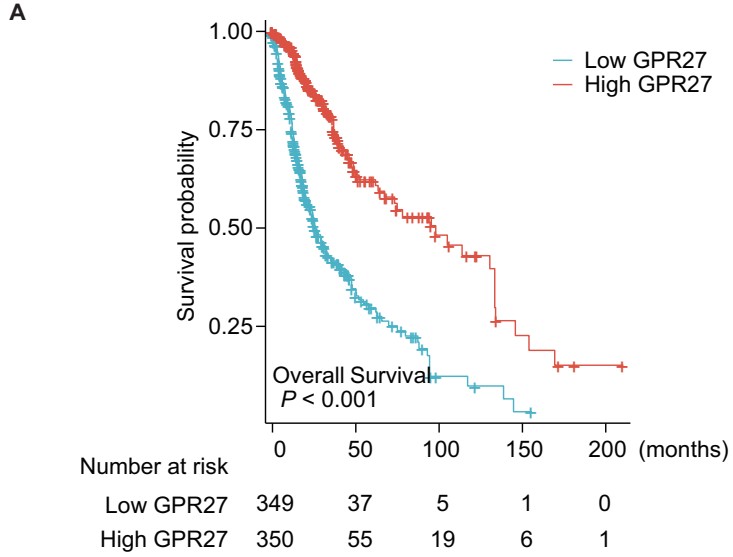

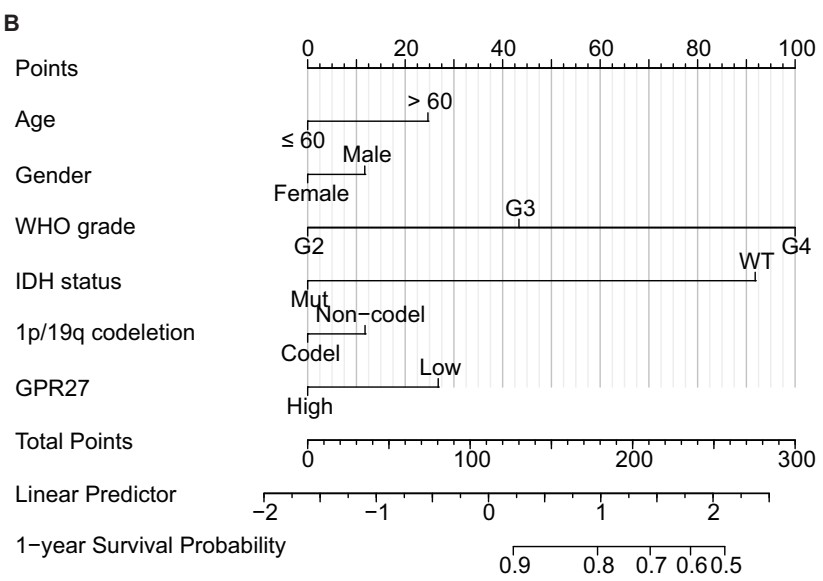

**Figure 2** **Prognostic significance of GPR27 on glioma patients.** (A) Kaplan–Meier survival analysis of glioma patients based on GPR27 expression level. Patients with lower GPR27 level had significantly worse prognosis compared to those with higher GPR27 level ($P < 0.001$). (B) A nomogram established based on the multivariate survival analysis. The variables in the nomogram included patients' age, gender, WHO grade, IDH status, 1p/19q co-deletion, and GPR27 expression level.

have significant implications for the development of novel therapeutic strategies for this devastating disease.

**Table 2  Univariate and multivariate Cox regression analyses of the overall survival of glioma patients.**

| Characteristics | Total (N) | Univariate analysis | | Multivariate analysis | |
|---|---|---|---|---|---|
| | | Hazard ratio (95% CI) | *P* value | Hazard ratio (95% CI) | *P* value |
| Age | 698 | | | | |
| ≤60 years old | 555 | Reference | | Reference | |
| >60 years old | 143 | 4.696 (3.620–6.093) | <0.001*** | 1.429 (1.046–1.952) | 0.025* |
| Gender | 698 | | | | |
| Female | 297 | Reference | | Reference | |
| Male | 401 | 1.250 (0.979–1.595) | 0.073 | 1.185 (0.903–1.553) | 0.220 |
| WHO grade | 636 | | | | |
| G2 | 223 | Reference | | Reference | |
| G3 | 245 | 2.967 (1.986–4.433) | <0.001*** | 1.873 (1.223–2.868) | 0.004** |
| G4 | 168 | 18.600 (12.448–27.794) | <0.001*** | 4.253 (2.525–7.162) | <0.001*** |
| IDH status | 688 | | | | |
| WT | 246 | Reference | | Reference | |
| Mutation | 442 | 0.116 (0.089–0.151) | <0.001*** | 0.265 (0.176–0.399) | <0.001*** |
| 1p/19q codeletion | 691 | | | | |
| Non-codel | 520 | Reference | | Reference | |
| Codel | 171 | 0.225 (0.147–0.346) | <0.001*** | 0.843 (0.494–1.440) | 0.533 |
| GPR27 | 698 | | | | |
| Low | 349 | Reference | | Reference | |
| High | 349 | 0.326 (0.251–0.423) | <0.001*** | 0.679 (0.486–0.947) | 0.023* |

Notes.
*$P<0.05$
**$P<0.01$
***$P<0.001$

### GPR27 is correlated with the immune cell infiltration in gliomas

The results of GSEA showed that the GPR27 has a negative association with macrophages ($R = -0.551$, $P < 0.001$), neutrophils ($R = -0.473$, $P < 0.001$), aDC ($R = -0.405$, $P < 0.001$), eosinophils ($R = -0.397$, $P < 0.001$), iDC cells ($R = -0.324$, $P < 0.001$), cytotoxic cells ($R = -0.286$, $P < 0.001$), *etc*. These findings indicate that GPR27 may play a crucial role in modulating the immune response in the tumor microenvironment. Oppositely, GPR27 shows positive associations with pDC cells ($R = 0.325$, $P < 0.001$), TFH cells ($R = 0.281$, $P = 0.003$), NK CD56bright cells ($R = 0.251$, $P < 0.001$), Tcm cells ($R = 0.214$, $P < 0.001$), TReg cells ($R = 0.190$, $P < 0.001$), *etc* (Figs. 4A–4C). These results suggest that GPR27 may play a significant role in promoting the differentiation and activation of these immune cells.

## DISCUSSIONS

The results presented in this study suggest that GPR27 expression levels are closely correlated with the disease status of glioma patients. Specifically, GPR27 expression was negatively correlated with WHO grade, and patients with IDH mutation or 1p/19q co-deletion exhibited higher GPR27 levels. Additionally, *in silico* analysis showed that lower GPR27 expression was correlated with higher death rates in glioma patients. In line

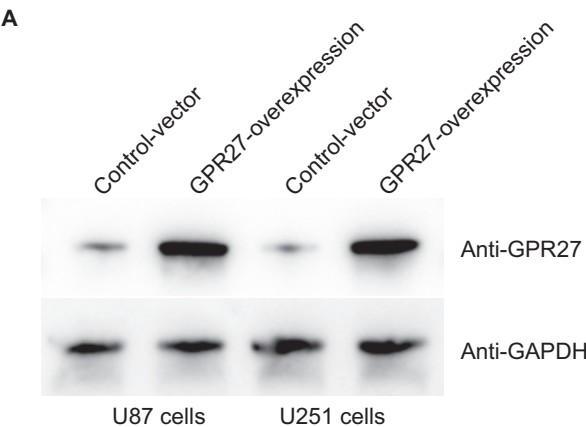

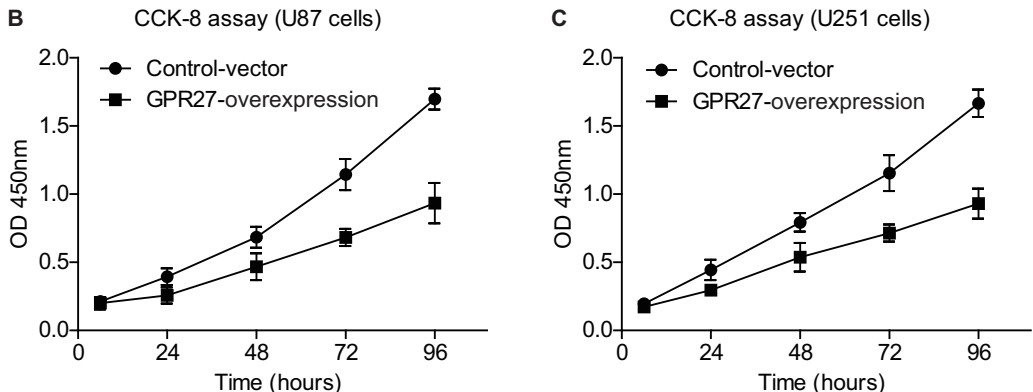

**Figure 3   GPR27-overexpression inhibit glioma progression.** (A) Western blot analysis confirmed the successful overexpression of GPR27 expression in U87 and U251 cell lines. (B) CCK-8 assay results indicated that overexpressing GPR27 significantly inhibited the proliferation of U87 cells compared to the control group. (C) CCK-8 assay results indicated that overexpressing GPR27 significantly inhibited the proliferation of U251 cells compared to the control group. Each experiment was repeated three times independently.

with these findings, the authors performed further analyses and showed that low GPR27 expression was associated with elder patients, higher-grade gliomas, and wild-type IDH status or non-codeletion of 1p/19q. Importantly, Kaplan–Meier survival analysis indicated that lower GPR27 expression was a significant independent predictor of poor overall survival in glioma patients. Taken together, clinical data analyses suggest that GPR27 may serve as a useful prognostic biomarker for glioma and may have implications for the development of novel therapeutic strategies.

Consistently, cellular experiments suggest that GPR27 plays a crucial role in the development and progression of glioma, as overexpressing GPR27 expression significantly inhibited the proliferation capacities of U87 and U251 cell lines. These findings are distinct with previous reports that have demonstrated that GPR27 is overexpressed in several types of cancer and promotes tumor growth and metastasis. For example, *Wang et al. (2022)* confirmed that GPR27 expression was upregulated in HCC tissues and cell

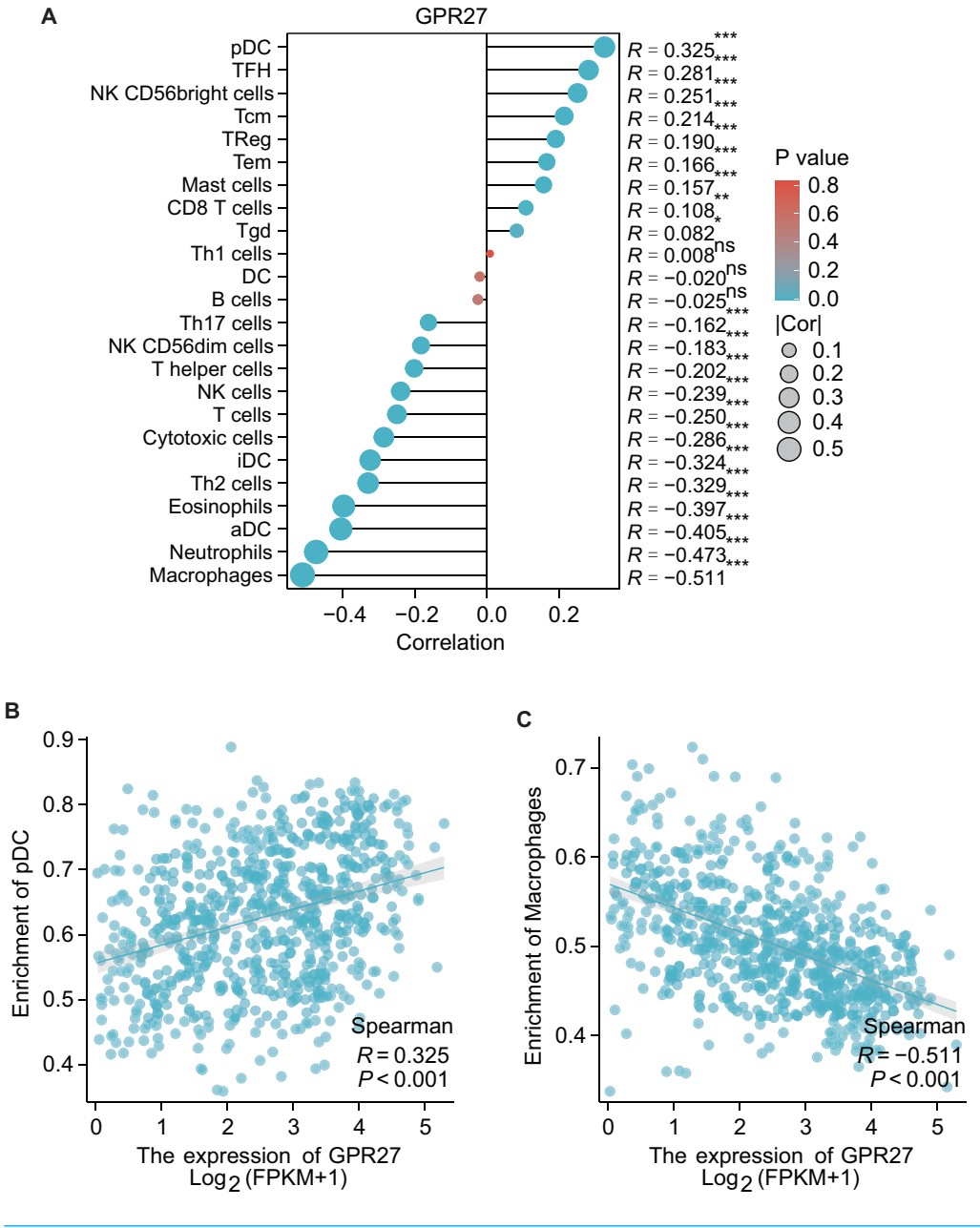

**Figure 4 The correlation between immune cell infiltration and GPR27 expression in glioma.** (A) Spearman analysis result showed the correlation between the infiltration of 24 types of immune cells and GPR27 expression in glioma tissues. (B) and (C) showed representative infiltration level ofpDC and macrophages in cells with different GPR27 level. DC, dendritic cells; pDC, plasmacytoid DC; aDC, activated DC; iDCs, immature DCs; NK, natural killer cells; Tgd, T gamma delta; TReg, regulatory T cells; Tem, T effector memory; Tcm, T central memory; Th1 cells, type 1 Th cells; Th2 cells, type 2 Th cells; Th17 cells, type 17 Th cells; TFH, T follicular helper.

lines. They then used cellular experiments to show that knockdown of GPR27 inhibited HCC cell proliferation, migration, and invasion. Further analysis revealed that GPR27 acted through the MAPK/ERK pathway to promote HCC progression (*Wang et al., 2022*). Besides the classical downstream G-protein signaling, GPR27 can also activate beta-arrestin-biased signaling pathways (*Dupuis et al., 2017*). Meanwhile, beta-arrestins had been well-acknowledged to be involved in malignancies including glioblastoma (*Lan et al., 2017*; *Bae et al., 2021*). Therefore, it is high likely that GPR27 may modulate glioma progression *via* distinct signaling pathways in different cancer types. Furthermore, the identification of GPR27 as a potential therapeutic target for glioma treatment is of significant clinical relevance, as there is a dire need for novel therapeutic strategies for this devastating disease. Further studies will be necessary to further explore its functional mechanisms.

The present study provides important insights underlying glioma development and progression. The observed negative correlation between GPR27 and various immune cell types suggests that GPR27 may play a role in immune evasion by glioma cells, and further investigations into the mechanism underlying this association could help to develop effective immunotherapeutic strategies (*Njonkou et al., 2022*; *Razavi et al., 2016*). Moreover, the identification of GPR27 as a potential therapeutic target could pave the way for the development of new drugs, which could offer a promising avenue for glioma treatment. However, more research is needed to fully understand the mechanisms underlying the role of GPR27 in glioma development and progression, as well as the potential therapeutic implications of targeting GPR27.

Our research has revealed notable connections between GPR27 and various immune cells, including pDC and macrophages. Specifically, data from Serena Pellegatta et al. indicate that administering pDC directly into tumors considerably extends survival in a murine model of glioma (*Pellegatta et al., 2010*). This study indicates that pDC within the tumor can enhance anti-cancer immune activity, primarily through modulating pro-immune cytokines, reducing Treg cells, and directly curbing tumor growth *via* TNF-$\alpha$. Given the observed positive link between GPR27 and pDC presence, it's plausible that the interaction between GPR27 and pDC might contribute, at least in part, to GPR27's tumor-fighting properties. Conversely, our findings indicate an inverse relationship between GPR27 and macrophage presence, suggesting that lower GPR27 levels could be associated with increased macrophage presence within the tumor environment. This is supported by findings from *Poon et al. (2017)*, who observed that glioblastoma-associated macrophages, when co-cultured alongside glioblastoma stem cells, enhanced the growth of glioblastoma cells. This underscores the potential of macrophages within the glioma environment to facilitate tumor growth.

The implications of these findings could be far-reaching in the field of cancer immunotherapy. Given the crucial role that the tumor microenvironment plays in cancer progression (*Barthel et al., 2022*), GPR27 could serve as a potential target for the development of novel cancer immunotherapies. By moduclating the expression of GPR27, it may be possible to adjust the immune response in the tumor microenvironment to promote an anti-tumor immune response. Alternatively, targeting GPR27 could be used to promote the differentiation and activation of immune cells that are positively associated

with GPR27, such as pDC cells and TReg cells. Overall, these findings provide important insights into the role of GPR27 in the modulation of the immune response in the tumor microenvironment and could have significant implications for the development of novel cancer immunotherapies.

The study has several limitations, such as a restricted sample size, which may not be representative of the entire population. Moreover, the study's outcomes may not be generalized to other regions due to its geographic confinement. The self-reported data may also be influenced by potential biases. Furthermore, the study's cross-sectional design prevents establishing a cause-and-effect relationship between variables.

## CONCLUSIONS

Taken together, our data highlights the potential clinical significance of GPR27 in gliomas. The findings indicate that GPR27 expression is negatively correlated with WHO grade. Additionally, lower GPR27 expression is associated with elder patients, higher-grade gliomas, and specimens with wild-type IDH status or non-codeletion of 1p/19q, and predicts worse prognosis. These results suggest that GPR27 may be a potential prognostic biomarker for glioma patients. However, further research is needed to fully understand the role of GPR27 in gliomas and to explore its potential as a therapeutic target.

### Funding
The authors received no funding for this work.

### Competing Interests
The authors declare there are no competing interests.

### Author Contributions
- Changcheng Cai conceived and designed the experiments, performed the experiments, prepared figures and/or tables, and approved the final draft.
- Libo Hu analyzed the data, authored or reviewed drafts of the article, and approved the final draft.
- Ke Wu analyzed the data, authored or reviewed drafts of the article, and approved the final draft.
- Yinggang Liu analyzed the data, authored or reviewed drafts of the article, and approved the final draft.

### Human Ethics
The following information was supplied relating to ethical approvals (i.e., approving body and any reference numbers):

The Ethics Committee of Suining Central Hospital requires no ethic approval and grants an exemption from informed consent for this public-database study

## Data Availability

The data is available at figshare: Liu, Yinggang (2023). Orginal data for GPR27 expression correlates with prognosis and tumor progression in gliomas. figshare. Dataset. Available at https://doi.org/10.6084/m9.figshare.24406603.v2.

## Supplemental Information

Supplemental information for this article can be found online at http://dx.doi.org/10.7717/peerj.17024#supplemental-information.

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
