# Peer review of "GPR27 expression correlates with prognosis and tumor progression in gliomas"

_PeerJ, doi:10.7717/peerj.17024_

## Round 0.1 · original submission · Major Revisions

Please accept my apologies for the delay in reviewing your manuscript. The main concern is that neither reviewer is convinced of the validity of your data and conclusions. Both recommend major revisions. The revised manuscript needs to fully address the concerns raised by the reviewers, particularly those related to the Western blotting data.

Reviewer 1 ·

Basic reporting

Glioma is a highly aggressive brain tumor with a poor prognosis, and the role of G protein-coupled receptor 27 (GPR27) in gliomas is not well understood. This study assessed GPR27 expression in gliomas using GTEx and TCGA datasets, finding significantly lower levels in glioma tissues compared to normal brain specimens. GPR27 expression correlated with disease status, WHO grade, and molecular markers such as IDH mutation and 1p/19q co-deletion. Lower GPR27 levels were associated with higher mortality. Cellular experiments confirmed that GPR27 inhibits glioma cell growth. The findings suggest that GPR27 could serve as a potential prognostic biomarker and therapeutic target in gliomas, urging further research to elucidate its signaling mechanism and clinical implications. While the study provides valuable insights, there are notable shortcomings in the methodological and conclusion sections that warrant attention.

Experimental design

1. In Figure 1A, the authors appear to combine data from GTEx and TCGA datasets. It is crucial to recognize that these datasets are derived from distinct cohorts employing different experimental methodologies. Consequently, direct comparisons between these datasets may not be appropriate.
2. The statement, "Moreover, in silico analysis revealed that lower GPR27 level was correlated with higher death rates (Figure 1E, P<0.001)," lacks sufficient detail. The authors should elaborate on the specific methods employed for the in-silico analysis to enhance the transparency of their findings.

Validity of the findings

1. Discrepancies regarding the number of glioma samples raise concerns. In the methods section, the authors state that they downloaded 689 glioma samples' RNA-seq data from TCGA (https://portal.gdc.cancer.gov/projects/TCGA-GBM), but this TCGA webpage only lists 167 samples with RNA-seq data. Additionally, in Table 1, the division of TCGA cohorts into high-GPR27 (n=350) and low-GPR27 (n=349) groups results in a total of 699, inconsistent with the previously mentioned 689.
2. The authors could enhance the clarity of Figure 2A by including the 'n' values for each time point beneath the Kaplan-Meier curve, providing a clearer understanding of the sample sizes at different intervals.
3. The methodology section requires additional details, especially concerning experimental procedures. For western blot experiments, it is essential to document the antibody used, its concentration, and the control protein. Furthermore, both western blot and CCK-8 experiments should include information on the number of replicates employed, contributing to the reproducibility of the study.
4. The authors should elucidate the rationale behind the observed changes in immune cells and their correlation with patient survival. A more comprehensive explanation of the relationship between immune cell alterations and clinical outcomes would strengthen the study's interpretation and significance.

Reviewer 2 ·

Basic reporting

The language could be improved for clarity as well.
Eg. “RNA expression information in the Fragments Per Kilobase per Million format (FPKM) as well as
78 related clinic-pathological data of glioma samples (n=689) and normal brain samples (n=1157)
79 were downloaded from TCGA (https://portal.gdc.cancer.gov/projects/TCGA-GBM) and GTEx
80 (https://www.gtexportal.org/home/).” Need to break down the sentence.

“Moreover, in silico analysis revealed that lower-GPR27 level was
128 correlated with higher death rates (Figure 1E, P<0.001) although its prognostic significance
129 remains further investigation.” Incorrect use of remains
“was replaced every two days, and cells were sub- seeded once they reach 70-80% confluence.” More grammatical mistake “reached”

Experimental design

Specifically, the article described used a knockdown approach of GPR27 with siRNAs targeting GPR27 or scramble siRNA as a control. However, the authors did not list the specific sequences or the design of these siRNA. This is a critical information to be included in the results section. Furthermore, I did not see the information of the antibody against GPR27 listed in the method section for doing the western blot. The method section did not mention how the western blot is conducted either. Given the variability of the antibody, the authors need to validate the specificity of the antibody used against GPR27. These all make the western blot uninterpretable in figure 3. Given what’s presented there, the author needs to explain what’s their staining control is in the bottom half of the panel. Also, why is the level of GPR27 lower in the control band in the top panel. The authors need to make it clear what they are showing in this figure.

Validity of the findings

see my above comment. I have issue with the current conclusion and validity of the data

Additional comments

1. The violin plot in Figure 1 could be refined for better visualization. Specifically, there is no need to show the rectangle indicating the quarter range. The way it is presented now, it is obscuring all the individual data points. I would just leave the few horizontal lines indicating 25%,50% and 75% value.

2. Define low and high level of GPR27 for figure 2. How do you define the high and low levels?

3. Give more explanation of what the CCK-8 experiment is

---

## Round 0.2 · Minor Revisions

Please address the remaining points raised by the reviewers.

Reviewer 1 ·

Basic reporting

In the revised manuscript, the authors address all my previous concerns and enhance the quality of the manuscript.
The only problem I found is: in table 1, the total cases for low GPR27 and high GPR27 are 349 and 350 respectively. However, in figure 2, the numbers are incorrectly both labeled as 349 at the baseline.

Experimental design

no comment

Validity of the findings

no comment

Reviewer 2 ·

Basic reporting

The main structure and findings of the manuscript stay the same.

Experimental design

The authors have addressed my points on the western blot and the results are clarified. In the method session, please include the source of where you get the plasmid or the submit the vector map as a supplementary information.

Validity of the findings

no comment

Additional comments

I have no further comments

---

## Round 0.3 · accepted · Accept

The authors addressed the reviewers' comments and now can proceed to publication.